# Query2doc: Query Expansion with Large Language Models

**Liang Wang** and **Nan Yang** and **Furu Wei**
Microsoft Research
{wangliang,nanya,fuwei}@microsoft.com

## Abstract

This paper introduces a simple yet effective query expansion approach, denoted as *query2doc*, to improve both sparse and dense retrieval systems. The proposed method first generates pseudo-documents by few-shot prompting large language models (LLMs), and then expands the query with generated pseudo-documents. LLMs are trained on web-scale text corpora and are adept at knowledge memorization. The pseudo-documents from LLMs often contain highly relevant information that can aid in query disambiguation and guide the retrievers. Experimental results demonstrate that *query2doc* boosts the performance of BM25 by 3% to 15% on ad-hoc IR datasets, such as MS-MARCO and TREC DL, without any model fine-tuning. Furthermore, our method also benefits state-of-the-art dense retrievers in terms of both in-domain and out-of-domain results.

## 1 Introduction

Information retrieval (IR) aims to locate relevant documents from a large corpus given a user issued query. It is a core component in modern search engines and researchers have invested for decades in this field. There are two mainstream paradigms for IR: lexical-based sparse retrieval, such as BM25, and embedding-based dense retrieval (Xiong et al., 2021; Qu et al., 2021). Although dense retrievers perform better when large amounts of labeled data are available (Karpukhin et al., 2020), BM25 remains competitive on out-of-domain datasets (Thakur et al., 2021).

Query expansion (Rocchio, 1971; Lavrenko and Croft, 2001) is a long-standing technique that rewrites the query based on pseudo-relevance feedback or external knowledge sources such as WordNet. For sparse retrieval, it can help bridge the lexical gap between the query and the documents. However, query expansion methods like RM3 (Lavrenko and Croft, 2001; Lv and Zhai,

2009) have only shown limited success on popular datasets (Campos et al., 2016), and most state-of-the-art dense retrievers do not adopt this technique. In the meantime, document expansion methods like doc2query (Nogueira et al., 2019) have proven to be effective for sparse retrieval.

In this paper, we demonstrate the effectiveness of LLMs (Brown et al., 2020) as query expansion models by generating pseudo-documents conditioned on few-shot prompts. Given that search queries are often short, ambiguous, or lack necessary background information, LLMs can provide relevant information to guide retrieval systems, as they memorize an enormous amount of knowledge and language patterns by pre-training on trillions of tokens.

Our proposed method, called *query2doc*, generates pseudo-documents by few-shot prompting LLMs and concatenates them with the original query to form a new query. This method is simple to implement and does not require any changes in training pipelines or model architectures, making it orthogonal to the progress in the field of LLMs and information retrieval. Future methods can easily build upon our query expansion framework.

For in-domain evaluation, we adopt the MS-MARCO passage ranking (Campos et al., 2016), TREC DL 2019 and 2020 datasets. Pseudo-documents are generated by prompting an improved version of GPT-3 *text-davinci-003* from OpenAI (Brown et al., 2020). Results show that *query2doc* substantially improves the off-the-shelf BM25 algorithm without fine-tuning any model, particularly for hard queries from the TREC DL track. Strong dense retrievers, including DPR (Karpukhin et al., 2020), SimLM (Wang et al., 2023), and E5 (Wang et al., 2022) also benefit from *query2doc*, although the gains tend to be diminishing when distilling from a strong cross-encoder based re-ranker. Experiments in zero-shot OOD settings demonstrate that

our method outperforms strong baselines on most datasets. Further analysis also reveals the importance of model scales: *query2doc* works best when combined with the most capable LLMs while small language models only provide marginal improvements over baselines. To aid reproduction, we release all the generations from *text-davinci-003* at https://huggingface.co/datasets/intfloat/query2doc_msmarco.

## 2 Method

Figure 1: Illustration of *query2doc* few-shot prompting. We omit some in-context examples for space reasons.

Given a query $q$, we employ few-shot prompting to generate a pseudo-document $d'$ as depicted in Figure 1. The prompt comprises a brief instruction *"Write a passage that answers the given query:"* and $k$ labeled pairs randomly sampled from a training set. We use $k = 4$ throughout this paper. Subsequently, we rewrite $q$ to a new query $q^+$ by concatenating with the pseudo-document $d'$. There are slight differences in the concatenation operation for sparse and dense retrievers, which we elaborate on in the following section.

**Sparse Retrieval** Since the query $q$ is typically much shorter than pseudo-documents, to balance the relative weights of the query and the pseudo-document, we boost the query term weights by repeating the query $n$ times before concatenating with the pseudo-document $d'$:

$$q^+ = \text{concat}(\{q\} \times \text{n},\ d') \qquad (1)$$

Here, "concat" denotes the string concatenation function. $q^+$ is used as the new query for BM25 retrieval. We find that $n = 5$ is a generally good value and do not tune it on a dataset basis.

**Dense Retrieval** The new query $q^+$ is a simple concatenation of the original query $q$ and the pseudo-document $d'$ separated by [SEP]:

$$q^+ = \text{concat}(q,\ [\text{SEP}],\ d') \qquad (2)$$

For training dense retrievers, several factors can influence the final performance, such as hard negative mining (Xiong et al., 2021), intermediate pre-training (Gao and Callan, 2021), and knowledge distillation from a cross-encoder based re-ranker (Qu et al., 2021). In this paper, we investigate two settings to gain a more comprehensive understanding of our method. The first setting is training DPR (Karpukhin et al., 2020) models initialized from BERT$_{\text{base}}$ with BM25 hard negatives only. The optimization objective is a standard contrastive loss:

$$L_{\text{cont}} = -\log \frac{e^{\mathbf{h}_q \cdot \mathbf{h}_d}}{e^{\mathbf{h}_q \cdot \mathbf{h}_d} + \sum_{d_i \in \mathbb{N}} e^{\mathbf{h}_q \cdot \mathbf{h}_{d_i}}} \qquad (3)$$

where $\mathbf{h}_q$ and $\mathbf{h}_d$ represent the embeddings for the query and document, respectively. $\mathbb{N}$ denotes the set of hard negatives.

The second setting is to build upon state-of-the-art dense retrievers and use KL divergence to distill from a cross-encoder teacher model.

$$\min \quad D_{\text{KL}}(p_{\text{ce}}, p_{\text{stu}}) + \alpha L_{\text{cont}} \qquad (4)$$

$p_{\text{ce}}$ and $p_{\text{stu}}$ are the probabilities from the cross-encoder and our student model, respectively. $\alpha$ is a coefficient to balance the distillation loss and contrastive loss.

**Comparison with Pseudo-relevance Feedback** Our proposed method is related to the classic method of pseudo-relevance feedback (PRF) (Lavrenko and Croft, 2001; Lv and Zhai, 2009). In conventional PRF, the feedback signals for query expansion come from the top-k documents obtained in the initial retrieval step, while our method prompts LLMs to generate pseudo-documents. Our method does not rely on the quality of the initial retrieval results, which are often noisy or irrelevant. Rather, it exploits cutting-edge LLMs to generate documents that are more likely to contain relevant terms.

| Method | Fine-tuning | MS MARCO dev | | | TREC DL 19 | TREC DL 20 |
|---|---|---|---|---|---|---|
| | | MRR@10 | R@50 | R@1k | nDCG@10 | nDCG@10 |
| **Sparse retrieval** | | | | | | |
| BM25 | ✗ | 18.4 | 58.5 | 85.7 | 51.2* | 47.7* |
| + query2doc | ✗ | $21.4^{+3.0}$ | $65.3^{+6.8}$ | $91.8^{+6.1}$ | $\mathbf{66.2}^{+15.0}$ | $\mathbf{62.9}^{+15.2}$ |
| BM25 + RM3 | ✗ | 15.8 | 56.7 | 86.4 | 52.2 | 47.4 |
| docT5query (Nogueira and Lin) | ✓ | **27.7** | **75.6** | **94.7** | 64.2 | - |
| **Dense retrieval w/o distillation** | | | | | | |
| ANCE (Xiong et al., 2021) | ✓ | 33.0 | - | 95.9 | 64.5 | 64.6 |
| HyDE (Gao et al., 2022) | ✗ | - | - | - | 61.3 | 57.9 |
| DPR$_{\text{bert-base}}$ (our impl.) | ✓ | 33.7 | 80.5 | 95.9 | 64.7 | 64.1 |
| + query2doc | ✓ | $\mathbf{35.1}^{+1.4}$ | $\mathbf{82.6}^{+2.1}$ | $\mathbf{97.2}^{+1.3}$ | $\mathbf{68.7}^{+4.0}$ | $\mathbf{67.1}^{+3.0}$ |
| **Dense retrieval w/ distillation** | | | | | | |
| RocketQAv2 (Ren et al., 2021) | ✓ | 38.8 | 86.2 | 98.1 | - | - |
| AR2 (Zhang et al., 2022) | ✓ | 39.5 | 87.8 | 98.6 | - | - |
| SimLM (Wang et al., 2023) | ✓ | 41.1 | 87.8 | 98.7 | 71.4 | 69.7 |
| + query2doc | ✓ | $\mathbf{41.5}^{+0.4}$ | $\mathbf{88.0}^{+0.2}$ | $\mathbf{98.8}^{+0.1}$ | $\mathbf{72.9}^{+1.5}$ | $\mathbf{71.6}^{+1.9}$ |
| E5$_{\text{base}}$ + KD (Wang et al., 2022) | ✓ | 40.7 | 87.6 | 98.6 | 74.3 | 70.7 |
| + query2doc | ✓ | $\mathbf{41.5}^{+0.8}$ | $\mathbf{88.1}^{+0.5}$ | $98.7^{+0.1}$ | $\mathbf{74.9}^{+0.6}$ | $\mathbf{72.5}^{+1.8}$ |

Table 1: Main results on the MS-MARCO passage ranking and TREC datasets. The "Fine-tuning" column indicates whether the method requires fine-tuning model on labeled data or not. *: our reproduction.

## 3 Experiments

### 3.1 Setup

**Evaluation Datasets** For in-domain evaluation, we utilize the MS-MARCO passage ranking (Campos et al., 2016), TREC DL 2019 (Craswell et al., 2020a) and 2020 (Craswell et al., 2020b) datasets. For zero-shot out-of-domain evaluation, we select five low-resource datasets from the BEIR benchmark (Thakur et al., 2021). The evaluation metrics include MRR@10, R@k (k $\in$ {50, 1k}), and nDCG@10.

**Hyperparameters** For sparse retrieval including BM25 and RM3, we adopt the default implementation from Pyserini (Lin et al., 2021). When training dense retrievers, we use mostly the same hyperparameters as SimLM (Wang et al., 2023), with the exception of increasing the maximum query length to 144 to include pseudo-documents. When prompting LLMs, we include 4 in-context examples and use the default temperature of 1 to sample at most 128 tokens. For further details, please refer to Appendix A.

### 3.2 Main Results

In Table 1, we list the results on the MS-MARCO passage ranking and TREC DL datasets. For sparse retrieval, "BM25 + query2doc" beats the BM25 baseline with over 15% improvements on TREC DL 2019 and 2020 datasets. Our manual inspection reveals that most queries from the TREC DL track

are long-tailed entity-centric queries, which benefit more from the exact lexical match. The traditional query expansion method RM3 only marginally improves the R@1k metric. Although the document expansion method docT5query achieves better numbers on the MS-MARCO dev set, it requires training a T5-based query generator with all the available labeled data, while "BM25 + query2doc" does not require any model fine-tuning.

For dense retrieval, the model variants that combine with query2doc also outperform the corresponding baselines on all metrics. However, the gain brought by query2doc tends to diminish when using intermediate pre-training or knowledge distillation from cross-encoder re-rankers, as shown by the "SimLM + query2doc" and "E5 + query2doc" results.

For zero-shot out-of-domain retrieval, the results are mixed as shown in Table 2. Entity-centric datasets like DBpedia see the largest improvements. On the NFCorpus and Scifact datasets, we observe a minor decrease in ranking quality. This is likely due to the distribution mismatch between training and evaluation.

## 4 Analysis

**Scaling up LLMs is Critical** For our proposed method, a question that naturally arises is: how does the model scale affect the quality of query expansion? Table 3 shows that the performance steadily improves as we go from the 1.3B model

|  | DBpedia | NFCorpus | Scifact | Trec-Covid | Touche2020 |
|---|---|---|---|---|---|
| BM25 | 31.3 | 32.5 | 66.5 | 65.6 | 36.7 |
| + query2doc | 37.0$^{+5.7}$ | 34.9$^{+2.4}$ | 68.6$^{+2.1}$ | 72.2$^{+6.6}$ | **39.8**$^{+3.1}$ |
| SimLM (Wang et al., 2023) | 34.9 | 32.7 | 62.4 | 55.0 | 18.9 |
| + query2doc | 38.3$^{+3.4}$ | 32.1$^{-0.6}$ | 59.5$^{-2.9}$ | 59.9$^{+4.9}$ | 25.6$^{+6.7}$ |
| E5$_{base}$ + KD (Wang et al., 2022) | 40.7 | 35.0 | **70.4** | 74.1 | 30.9 |
| + query2doc | **42.4**$^{+1.7}$ | **35.2**$^{+0.2}$ | 67.5$^{-2.9}$ | **75.1**$^{+1.0}$ | 31.7$^{+0.8}$ |

Table 2: Zero-shot out-of-domain results on 5 low-resource datasets from the BEIR benchmark (Thakur et al., 2021). The reported numbers are nDCG@10. For a fair comparison, the in-context examples for prompting LLMs come from the MS-MARCO training set.

|  | # params | TREC 19 | TREC 20 |
|---|---|---|---|
| BM25 | - | 51.2 | 47.7 |
| w/ babbage | 1.3B | 52.0 | 50.2 |
| w/ curie | 6.7B | 55.1 | 50.1 |
| w/ davinci-001 | 175B | 63.5 | 58.2 |
| w/ davinci-003 | 175B | 66.2 | 62.9 |
| w/ gpt-4 | - | **69.2** | **64.5** |

Table 3: Query expansion with different model sizes. Even though GPT-4 performs best, we are unable to apply it in the main experiments due to quota limits.

to 175B models. Empirically, the texts generated by smaller language models tend to be shorter and contain more factual errors. Also, the "davinci-003" model outperforms its earlier version "davinci-001" by using better training data and improved instruction tuning. The recently released GPT-4 (OpenAI, 2023) achieves the best results.

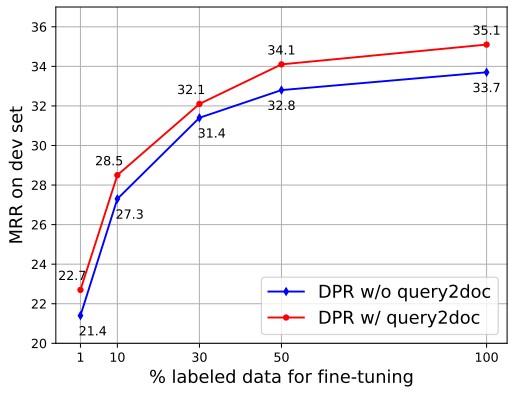

Figure 2: MRR on MS-MARCO dev set w.r.t the percentage of labeled data used for fine-tuning.

**Performance Gains are Consistent across Data Scales** Figure 2 presents a comparison between two variants of DPR models, which differ in the amount of labeled data used. The results show that the "DPR + query2doc" variant consistently outperforms the DPR baseline by approximately

1%, regardless of the amount of data used for fine-tuning. This observation highlights that our contribution is orthogonal to the continual scaling up of supervision signals.

|  | TREC 19 | TREC 20 |
|---|---|---|
| BM25 + query2doc | **66.2** | **62.9** |
| w/ query only | 51.2 | 47.7 |
| w/ pseudo-doc only | 48.7 | 44.5 |

Table 4: Using the concatenation of the original query and the generated pseudo-documents perform substantially better.

**How to Use Pseudo-documents** In this paper, we concatenate the original query and pseudo-documents as the new query. Alternatively, one can solely use the pseudo-documents, as done in the approach of HyDE (Gao et al., 2022). The results presented in Table 4 demonstrate that the original query and pseudo-documents are complementary, and their combination leads to substantially better performance in sparse retrieval.

**Case Analysis** In Table 5, we show two queries along with their respective pseudo-documents and groundtruth. The pseudo-documents, which are generated by LLMs, offer detailed and mostly accurate information, thereby reducing the lexical mismatch between the query and documents. In some cases, the pseudo-documents are sufficient to meet the user's information needs, rendering the retrieval step unnecessary. However, it is worth noting that the LLM generations may contain factual errors. For instance, in the second query, the theme song "It's a Jungle Out There" was used as of season two in 2003, not 2002 [1]. Although such errors may appear subtle and difficult to verify, they pose a significant challenge to building trustworthy systems using LLMs.

[1] Refer to https://en.wikipedia.org/wiki/It's_a_Jungle_Out_There_(song)

| query | who killed nicholas ii of russia |
|---|---|
| LLM generation | Nicholas II of Russia, the **last Tsar** of Russia, was executed by **Bolshevik revolutionaries** on July 17th, **1918**, along with his wife Alexandra and **their five children**. The execution was ordered by Vladimir **Lenin**, the leader of the Bolshevik Party. The actual execution was carried out by a firing squad of Bolshevik troops, and the bodies were later buried in a mass grave... |
| Groundtruth | (1868-**1918**). Nicholas II was the **last** emperor, or **tsar**, of Russia, serving from 1894 to 1917. Nicholas, his wife, and **their five children** were killed by the **Bolsheviks**, members of a **revolutionary** group (led by **Lenin**) who seized control of the government in Russia during the October Revolution (see Russian Revolution) of 1917. |
| query | who sings monk theme song |
| LLM generation | The theme song for the television show Monk is entitled "**It's a Jungle Out There**" and is sung by American singer-songwriter **Randy Newman**. The song was written specifically for the show, and *it has been used as the theme song since the series premiered in 2002*. It has been praised by critics and fans alike and is often regarded as one of the best theme songs in television history. |
| Groundtruth | exists and is an alternate of. The Monk theme song is **It's a Jungle Out There** by **Randy Newman**. The Monk theme song is **It's a Jungle Out There** by **Randy Newman**. |

Table 5: Examples from the TREC DL 2020 dataset. Bold texts are the overlapping words between groundtruth and pseudo-documents generated from LLMs. The italicized red sentence demonstrates a factual error in language model generations.

## 5 Related Work

**Query Expansion and Document Expansion** are two classical techniques to improve retrieval quality, particularly for sparse retrieval systems. Both techniques aim to minimize the lexical gap between the query and the documents. Query expansion typically involves rewriting the query based on relevance feedback (Lavrenko and Croft, 2001; Rocchio, 1971) or lexical resources such as Word-Net (Miller, 1992). In cases where labels are not available, the top-k retrieved documents can serve as pseudo-relevance feedback signals (Lv and Zhai, 2009). Liu et al. fine-tunes an encoder-decoder model to generate contextual clues.

In contrast, document expansion enriches the document representation by appending additional relevant terms. Doc2query (Nogueira et al., 2019) trains a seq2seq model to predict pseudo-queries based on documents and then adds generated pseudo-queries to the document index. Learned sparse retrieval models such as SPLADE (Formal et al., 2021) and uniCOIL (Lin and Ma, 2021) also learn document term weighting in an end-to-end fashion. However, most state-of-the-art dense retrievers (Ren et al., 2021; Wang et al., 2023) do not adopt any expansion techniques. Our paper demonstrates that strong dense retrievers also benefit from query expansion using LLMs.

**Large Language Models (LLMs)** such as GPT-3 (Brown et al., 2020), PaLM (Chowdhery et al., 2022), and LLaMA (Touvron et al., 2023) are trained on trillions of tokens with billions of parameters, exhibiting unparalleled generalization ability across various tasks. LLMs can follow instructions in a zero-shot manner or conduct in-context learning through few-shot prompting. Labeling a few high-quality examples only requires minimal human effort. In this paper, we employ few-shot prompting to generate pseudo-documents from a given query. A closely related recent work HyDE (Gao et al., 2022) instead focuses on the zero-shot setting and uses embeddings of the pseudo-documents for similarity search. HyDE implicitly assumes that the groundtruth document and pseudo-documents express the same semantics in different words, which may not hold for some queries. In the field of question answering, RECITE (Sun et al., 2022) and GENREAD (Yu et al., 2022) demonstrate that LLMs are powerful context generators and can encode abundant factual knowledge. However, as our analysis shows, LLMs can sometimes generate false claims, hindering their practical application in critical areas.

## 6 Conclusion

This paper presents a simple method *query2doc* to leverage LLMs for query expansion. It first prompts LLMs with few-shot examples to generate pseudo-documents and then integrates with existing sparse or dense retrievers by augmenting queries with generated pseudo-documents. The underlying motivation is to distill the LLMs through prompting. Despite its simplicity, empirical evaluations demonstrate consistent improvements across various retrieval models and datasets.

## Limitations

|          | LLM call | Index search |
|----------|----------|--------------|
| BM25     | -        | 16ms         |
| + query2doc | >2000ms | 177ms      |

Table 6: Latency analysis for retrieval systems with our proposed query2doc. We retrieve the top 100 results for MS-MARCO dev queries with a single thread and then average over all the queries. The latency for LLM API calls depends on server load and is difficult to precisely measure.

An apparent limitation is the efficiency of retrieval. Our method requires running inference with LLMs which can be considerably slower due to the token-by-token autoregressive decoding. Moreover, with query2doc, searching the inverted index also becomes slower as the number of query terms increases after expansion. This is supported by the benchmarking results in Table 6. Real-world deployment of our method should take these factors into consideration.

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

## A  Implementation Details

|  | DPR | w/ distillation |
|---|---|---|
| learning rate | $2 \times 10^{-5}$ | $3 \times 10^{-5}$ |
| PLM | BERT$_{base}$ | SimLM / E5$_{base-unsup}$ |
| # of GPUs | 4 | 4 |
| warmup steps | 1000 | 1000 |
| batch size | 64 | 64 |
| epoch | 3 | 6 |
| $\alpha$ | n.a. | 0.2 |
| negatives depth | 1000 | 200 |
| query length | 144 | 144 |
| passage length | 144 | 144 |
| # of negatives | 15 | 23 |

Table 7: Hyper-parameters for training dense retrievers on MS-MARCO passage ranking dataset.

For dense retrieval experiments in Table 1, we list the hyperparameters in Table 7. When training dense retrievers with distillation from cross-encoder, we use the same teacher score released by Wang et al.. The SimLM and E5 checkpoints for initialization are publicly available at `https://huggingface.co/intfloat/simlm-base-msmarco` and `https://huggingface.co/intfloat/e5-base-unsupervised`. To compute the text embeddings, we utilize the [CLS] vector for SimLM and mean pooling for E5. This makes sure that the pooling mechanisms remain consistent between intermediate pre-training and fine-tuning. The training and evaluation of a dense retriever take less than 10 hours to finish.

When prompting LLMs, we include 4 in-context examples from the MS-MARCO training set. To increase prompt diversity, we randomly select 4 examples for each API call. A complete prompt is shown in Table 11. On the budget side, we make about 550k API calls to OpenAI's service, which costs nearly 5k dollars. Most API calls are used to generate pseudo-documents for the training queries.

For GPT-4 prompting, we find that it has a tendency to ask for clarification instead of directly generating the pseudo-documents. To mitigate this issue, we set the system message to "*You are asked to write a passage that answers the given query. Do not ask the user for further clarification.*".

Regarding out-of-domain evaluations on DBpedia (Hasibi et al., 2017), NFCorpus (Boteva et al., 2016), Scifact (Wadden et al., 2020), Trec-Covid (Voorhees et al., 2021), and Touche2020 (Bondarenko et al., 2022), SimLM's results are based on the released checkpoint by Wang et al..

For ablation experiments in Figure 2, we fine-tune for 40 epochs or 18k steps, whichever is reached first.

## B  Exploring Other Prompting Strategies

|  | DL 2019 | DL 2020 |
|---|---|---|
| Query2doc | **69.2** | 64.5 |
| + iterative prompt | 68.6 | **64.8** |

Table 8: GPT-4 with iterative prompting to self-verify the generated pseudo-documents.

Instead of generating pseudo-documents in one round, recent work (Press et al., 2022) proposes to iteratively prompt the LLM to improve the generation quality. We explore this intuition by asking GPT-4 to rewrite its own generated pseudo-documents with the following prompt template:

*You are asked to rewrite the passage that potentially answers the given query. You should only correct the factual errors in the passage, do not ask for clarification or make unnecessary changes.*

*Query: {{query}}*

*# Begin of passage*
*{{passage}}*
*# End of passage*

Empirically, we find that GPT-4 makes very few changes to the generated pseudo-documents, which suggests that the pseudo-documents are already of high quality or GPT-4 is not capable of correcting its own errors. The results are shown in Table 8.

| | |
|---|---|
| query | who sings hey good looking |
| LLM generation | Hey Good Lookin' was originally **written and recorded** by the country music singer **Hank Williams** in **1951**. The song has since **been covered** by several other **artists** and bands, including Jimmy Buffett, Conway Twitty, and Willie Nelson. |
| Groundtruth | Hey, Good Lookin'. (1951) Hey, Good Lookin' is a **1951** song **written and recorded** by **Hank Williams**, and his version was inducted into the Grammy Hall of Fame in 2001. Since its original 1951 recording it has **been covered** by a variety of **artists**. |
| query | trumbull marriott fax number |
| LLM generation | The fax number for the Trumbull Marriott Shelton is **203-378**-*4444*. |
| Groundtruth | Business name: Trumbull Marriott Merritt Parkway; Address: 180 Hawley Lane Trumbull, Connecticut 06611; Phone number: 203-378-4958; Fax number: **203-378**-1400; Business hours: 24; Credit cards accepted: Yes; Number of employees: 10-19; Map: |
| query | what is hra and hsa |
| LLM generation | HRA stands for Health Reimbursement Arrangement. It is a type of benefit plan where an **employer reimburses** employees for qualifying **medical expenses**. It also allows employers to set aside money tax-free to reimburse **employees** for medical expenses that are not covered under the employer's **health insurance** plan. HSA stands for Health Savings Account. It is a savings account that is used in conjunction with a qualified **High Deductible Health Plan** (**HDHP**). An HSA enables an individual to pay for current **health** expenses and save for future health expenses on a **tax**-free basis. Contributions to the account are tax |
| Groundtruth | The key difference between HRA and HSA is that HRA is an **employer** funded health benefit plan that **reimburses** for **medical expenses** including personal **health insurance** policy premiums of **employees** whereas HSA is also a **tax**-advantaged **health** benefit plan exclusively available to taxpayers in the United States who are enrolled in a **High-Deductible Health Plan** (**HDHP**). |

Table 9: More examples of LLM generations. The format is the same as in Table 5.

| | DL 2019 | DL 2020 |
|---|---|---|
| Average | 64.8 | 60.9 |
| Std dev. | ±1.14 | ±1.63 |

Table 10: Sparse retrieval results of *query2doc* across 3 random runs. The randomness comes from the selection of few-shot examples and the auto-regressive sampling of LLMs.

## C   Results Across Multiple Runs

In our method, there are two sources of randomness: the selection of few-shot examples and the auto-regressive top-p sampling of LLMs. To quantify the variance of our method, we report the average and standard deviation of sparse retrieval results across 3 random runs in Table 10. One possible improvement is to select few-shot examples based on semantic similarity to the query. We leave this for future work.

| | **Write a passage that answers the given query:** |
|---|---|
| | **Query:** what state is this zip code 85282 |
| | **Passage:** Welcome to TEMPE, AZ 85282. 85282 is a rural zip code in Tempe, Arizona. The population is primarily white, and mostly single. At $200,200 the average home value here is a bit higher than average for the Phoenix-Mesa-Scottsdale metro area, so this probably isn't the place to look for housing bargains.5282 Zip code is located in the Mountain time zone at 33 degrees latitude (Fun Fact: this is the same latitude as Damascus, Syria!) and -112 degrees longitude. |
| | **Query:** why is gibbs model of reflection good |
| | **Passage:** In this reflection, I am going to use Gibbs (1988) Reflective Cycle. This model is a recognised framework for my reflection. Gibbs (1988) consists of six stages to complete one cycle which is able to improve my nursing practice continuously and learning from the experience for better practice in the future.n conclusion of my reflective assignment, I mention the model that I chose, Gibbs (1988) Reflective Cycle as my framework of my reflective. I state the reasons why I am choosing the model as well as some discussion on the important of doing reflection in nursing practice. |
| prompts | **Query:** what does a thousand pardons means |
| | **Passage:** Oh, that's all right, that's all right, give us a rest; never mind about the direction, hang the direction - I beg pardon, I beg a thousand pardons, I am not well to-day; pay no attention when I soliloquize, it is an old habit, an old, bad habit, and hard to get rid of when one's digestion is all disordered with eating food that was raised forever and ever before he was born; good land! a man can't keep his functions regular on spring chickens thirteen hundred years old. |
| | **Query:** what is a macro warning |
| | **Passage:** Macro virus warning appears when no macros exist in the file in Word. When you open a Microsoft Word 2002 document or template, you may receive the following macro virus warning, even though the document or template does not contain macros: C:\<path>\<file name>contains macros. Macros may contain viruses. |
| | **Query:** when was pokemon green released |
| | **Passage:** |
| output | Pokemon Green was released in Japan on February 27th, 1996. It was the first in the Pokemon series of games and served as the basis for Pokemon Red and Blue, which were released in the US in 1998. The original Pokemon Green remains a beloved classic among fans of the series. |

Table 11: The full prompt used for the example in Figure 1.