# OpenReview forum: "Query2doc: Query Expansion with Large Language Models"
_EMNLP/2023/Conference — EMNLP 2023 Main_

### Official Review · Reviewer_VHFs · 2023-08-02

**Soundness:** 3

**Excitement:**

3: Ambivalent: It has merits (e.g., it reports state-of-the-art results, the idea is nice), but there are key weaknesses (e.g., it describes incremental work), and it can significantly benefit from another round of revision. However, I won't object to accepting it if my co-reviewers champion it.

**Paper Topic And Main Contributions:**

This paper is about a query expansion approach called query2doc, which aims to improve both sparse and dense retrieval systems in information retrieval. The paper addresses the problem of query disambiguation and how to guide the retrievers to improve performance on ad-hoc IR datasets. The main contribution of this paper is the introduction of query2doc, which generates pseudo-documents through large language models to aid in query disambiguation and guide the retrievers. The paper shows that query2doc can significantly boost the performance of retrieval systems, particularly on ad-hoc IR datasets. This paper makes a computationally-aided linguistic analysis contribution by proposing a new approach to query expansion that leverages large language models. The paper also provides publicly available software for query2doc.

**Reasons To Accept:**

1. The paper proposes an approach to query expansion, query2doc, that can significantly improve the performance of both sparse and dense retrieval systems on ad-hoc IR datasets.

2. The paper provides experimental results that demonstrate the effectiveness of query2doc and its benefits to state-of-the-art dense retrievers.

3. The paper makes a computationally-aided linguistic analysis contribution by proposing a new approach to query expansion that leverages large language models.

**Reasons To Reject:**

1. The proposed method is highly empirical and lacks theoretical justification, particularly in the case of repeating q for n times for sparse retrieval. It is empirical in the sense that the methodology does not appear to be able to generalize to other search tasks/datasets.

2. The evaluation does not compare the performance of dense retrievers using query2doc with other means of query/doc expansion, such as DPR/ES+Doc2Query.

3. The proposed approach prompts the LLM to make a guess about what a relevant document looks like, which could be incorrect and lead to degraded retrieval effectiveness. It might be useful to add a validation step that asks the LLM whether the generated passage answers the query. Additionally, a failure analysis should be included to allow deeper understanding of the limitations of the proposed approach. (Update: the case study is actually at the very end of the appendix.)

**Reproducibility:**

3: Could reproduce the results with some difficulty. The settings of parameters are underspecified or subjectively determined; the training/evaluation data are not widely available.

**Reviewer Confidence:**

4: Quite sure. I tried to check the important points carefully. It's unlikely, though conceivable, that I missed something that should affect my ratings.

---

> ### Author Rebuttal · Authors · 2023-08-28
>
> Thanks for your valuable feedback!
>
> **Q1: The proposed method is highly empirical and lacks theoretical justification, particularly in the case of repeating q for n times for sparse retrieval.**
>
> We appreciate your feedback and acknowledge that this paper is mostly empirical, aligning with the theme of the EMNLP conference which focuses on empirical methods in NLP.
>
> In regard to repeating q for n times for sparse retrieval, we would like to clarify that this is a form of term weighting. The BM25 matching score is a sum over all query terms. However, the generated pseudo-documents are often much longer than the original query, causing the BM25 score of the original query to be overwhelmed by the score of the pseudo-document. To balance their relative weights, we repeat the original query for n times. We will elaborate on this decision in the revised version.
>
> **Q2: The evaluation does not compare the performance of dense retrievers using query2doc with other means of query/doc expansion, such as DPR/ES+Doc2Query.**
>
> In terms of the evaluation, we aim to demonstrate that existing dense retrievers can be improved by using our query expansion techniques while keeping all other factors unchanged. This is an orthogonal contribution from document expansion, which can also be applied to further boost performance.
>
> **Q3: It might be useful to add a validation step that asks the LLM whether the generated passage answers the query. Additionally, a failure analysis should be included to allow deeper understanding of the limitations of the proposed approach.**
>
> We agree that asking the LLM to verify its own generations is a promising idea to explore, and we will add this part of analysis in the future revision.
>
> Regarding the failure analysis, we have included several cases in Table 7 and 8 in the Appendix, demonstrating that LLMs may hallucinate facts in the generated pseudo-documents, which can have a negative impact on retrieval systems. However, based on our manual inspection, such cases are relatively rare for LLMs like davinci-003 / gpt-4. Thank you for bringing this up.

---

### Official Review · Reviewer_hWUX · 2023-08-03

**Soundness:** 4

**Excitement:**

4: Strong: This paper deepens the understanding of some phenomenon or lowers the barriers to an existing research direction.

**Paper Topic And Main Contributions:**

In this paper, the authors use LLMs for the task of query expansion.
The authors propose to generate pseudo-relevant documents using an LLM and concatenate them to the original query. They experiment with both sparse and dense retrievers, and report strong performance for both of them.
The proposed method is evaluated on three standard datasets and is compared to several methods of different types.
Finally, the authors analyze the effect of model and labeled data size on query expansion performance, and found positive correlations for both.

**Reasons To Accept:**

- Using LLMs as proposed in this paper is a straightforward step to improve the quality of query expansion.
- The paper includes an interesting analysis section containing experiments for assessing the effect of model size and labeled data size on query expansion performance.
- The proposed method is evaluated on three standard datasets and is compared to several methods of different types - and shows strong performance.

**Reasons To Reject:**

- I didn't see any details about the computational complexity / budget used.
- I would prefer to see additional standard metrics in the main evaluation (Table 1).
- Statistical significance tests are not reported.

**Reproducibility:**

4: Could mostly reproduce the results, but there may be some variation because of sample variance or minor variations in their interpretation of the protocol or method.

**Reviewer Confidence:**

3: Pretty sure, but there's a chance I missed something. Although I have a good feel for this area in general, I did not carefully check the paper's details, e.g., the math, experimental design, or novelty.

---

> ### Author Rebuttal · Authors · 2023-08-28
>
> Thanks for your valuable review! We are thrilled to hear that you find our work interesting.
>
> **Q1: I didn't see any details about the computational complexity / budget used.**
>
> Thanks for pointing this out.
>
> For computational complexity, we provide the information on the hyperparameters and the number of GPUs used in Table 6. Training and evaluation of a dense retriever take less than 10 hours to finish.
>
> On the budget side, we make about 550k API calls to OpenAI's service, which costs nearly 5k dollars. Most API calls are used to generate pseudo-documents for the training queries.
>
> We will add these details in the future version of the paper.
>
> **Q2: I would prefer to see additional standard metrics in the main evaluation (Table 1).**
>
> For space reasons, we list the most widely adopted metrics for the corresponding datasets in Table 1. We find that other metrics such as MAP or Recall with different thresholds correlate with these main metrics quite well. We will consider adding additional metrics in the Appendix part.
>
> **Q3: Statistical significance tests are not reported.**
>
> We will update the paper numbers by averaging over different random runs and report significance tests whenever applicable to ensure the robustness and reliability of our results.
>
> Thank you once again for your feedback.

---

### Official Review · Reviewer_dbnL · 2023-08-04

**Soundness:** 4

**Excitement:**

4: Strong: This paper deepens the understanding of some phenomenon or lowers the barriers to an existing research direction.

**Paper Topic And Main Contributions:**

This paper discusses a simple query expansion approach using LLM. It is an incremental work compared to the HyDE model. The only difference between the two is that query2doc combines the original query with the generated pseudo-document, while HyDE only uses the generated document as the query to retrieve.

However, in my opinion, the contribution of this work mainly comes from experimentation. While HyDE only experimented with Contriever, this paper experiments with multiple retrieval models, including sparse and dense ones, which demonstrates the robustness and generalizability of the work. I am willing to vote for this paper based on its completeness of the experimentation, which is often less than ideal in retrieval papers submitted to *CL venues.

As a minor comment, the authors mentioned that the query2doc approach could be considered a pseudo-relevance feedback approach, which is, in fact, not the case. The proposed method is a query expansion technique that does not use the top-ranked documents as feedback signals. PRF is a set of query modification approaches that blindly assumes the top-ranked documents are relevant and modifies(often expands) the query based on them. Despite the similarity, query2doc should not be considered a PRF approach. However, the comparison is still a good one since both of them modify the query.

In summary, the authors provided a good set of experiments to demonstrate the generalizability and robustness of the proposed method. Despite the limited technological novelty in the modeling part, the experimentation is good. Therefore, I vote for acceptance for the paper.

**Reasons To Accept:**

Simple and effective query expansion model using LLM with a good set of experimentation.

**Reasons To Reject:**

Limited novelty in modeling.

**Reproducibility:**

4: Could mostly reproduce the results, but there may be some variation because of sample variance or minor variations in their interpretation of the protocol or method.

**Reviewer Confidence:**

4: Quite sure. I tried to check the important points carefully. It's unlikely, though conceivable, that I missed something that should affect my ratings.

**Typos Grammar Style And Presentation Improvements:**

- doc2query itself is a document expansion technique instead of a retrieval model. Since it uses BM25 underneath, it should be listed under BM25 in Table 1.

---

> ### Author Rebuttal · Authors · 2023-08-28
>
> Thanks for your valuable comments! We are encouraged to learn that you are impressed by the comprehensive experiments presented in our paper.
>
> **Q1: On limited novelty in modeling**
>
> While we did not introduce any new modeling architecture or training methods, our paper's main contribution lies in demonstrating through empirical evidence that LLMs can substantially enhance retrieval systems via query expansion. We believe this is an important finding that can have a big impact on the field, and we urge for the adoption of simple yet effective methods that can enhance retrieval systems.
>
> **Q2: On the discussion of query2doc as a Pseudo-Relevance Feedback method**
>
> We agree with the reviewer that query2doc does not strictly align with traditional PRF methods, as query2doc does not have an initial retrieval step to find top-ranked documents. Instead, the pseudo-documents in query2doc come from prompting off-the-shelf LLMs. Despite this difference, they share quite some similarities, therefore we include this comparison to help readers better understand this field.
>
> We are also thankful for the reviewer's helpful suggestions on presentation improvement.

---

### Meta-Review · Area_Chair_qSFY · 2023-09-19

**Recommendation:** 4

**Metareview:**

This paper proposes Query2doc to generate a potential relevant passage with LLMs and uses it to expand the query. The experiments show the effectiveness of the approach.
The experiments confirm the usefulness of LLMs for improving ad hoc IR in this way. It can inspire other researchers to work in a similar direction in the future.

---

### Decision · Program_Chairs · 2023-10-07

**Decision:**

Accept-Main

**Comment:**

This paper proposes Query2doc to generate a potential relevant passage with LLMs and uses it to expand the query. The experiments show the effectiveness of the approach.
The experiments confirm the usefulness of LLMs for improving ad hoc IR in this way. It can inspire other researchers to work in a similar direction in the future.